# A Polling-Based Transmission Scheme Using a Network Traffic Uniformity Metric for Industrial IoT Applications

**DOI:** 10.3390/s19010187

**Published:** 2019-01-07

**Authors:** Yuichi Igarashi, Ryo Nakano, Naoki Wakamiya

**Affiliations:** 1Center for Technology Innovation, Research & Development Group, Hitachi, Ltd., 1-280, Higashi-koigakubo Kokubunji-shi, Tokyo 185-8601, Japan; ryo.nakano.xd@hitachi.com; 2Graduate School of Information Science and Technology, Osaka University, Osaka 565-0871, Japan; wakamiya@ist.osaka-u.ac.jp

**Keywords:** the industrial IoT, polling-based communication, scheduling

## Abstract

The Industrial Internet of Things (IIoT) applications are required to provide precise measurement functions as feedback for controlling devices. The applications traditionally use polling-based communication protocols. However, in polling-based communication over current industrial wireless network protocols such as ISA100.11a, WirelessHART have difficulty in realizing both scheduled periodic data collection at high success ratio and unpredictable on-demand communications with short latency. In this paper, a polling-based transmission scheme using a network traffic uniformity metric is proposed for IIoT applications. In the proposed scheme, a center node controls the transmission timing of all polling-based communication in accordance with a schedule that is determined by a Genetic Algorithm. Communication of both periodic and unpredictable on-demand data collection are uniformly assigned to solve the above difficulties in the schedule. Simulation results show that network traffic is generated uniformly and a center node can collect periodic data from nodes at high success ratio. The average success probability of periodical data collection is 97.4% and the lowest probability is 95.2%.

## 1. Introduction

The Internet of Things (IoT) is a network of intelligent computers, sensors, and objects that collect, share, and analyze huge amounts of data [1,2,3]. Industrial applications and social infrastructure applications such as PA (Process Automation), DA ( Distribution Automation), AMI (Advanced Metering Infrastructure) and predictive maintenance recently use IoT technologies to improve production efficiency, ensure optimal resource consumption, and operate systems more efficiently and economically [4,5,6].

In DAs, a SCADA (Supervisory Control And Data Acquisition) system performs operations like bus voltage control, bus load balancing, circulating current control, overload control, transformer fault protection, and bus fault protection [7]. In this system, measurements are made by end devices called as Remote Terminal Units in the real field and then data are transferred to a central device called as a SCADA center so that complete process or manufacturing information can be provided remotely. A SCADA center acquires data through a process called polling to simplify communications between a central device and end devices.

In a typical AMI, DLMS/COSEM (Device Language Message Specification/Companion Specification for Energy Metering) [8] is used at the application layer. It is responsible for polling smart meters connected to the network and for sending the retrieved data to the management system from smart meters.

Both AMI and DA use polling schemes to exchange messages between a central device and end devices. In addition, a real-time system such as DA or PA reacts or responds to the applications within a fixed amount of time in order to avoid system failures. Both processing and reacting should be done within a pre-determined deadline that includes communication latency. Traditional PA systems use data collection protocols like Modbus that were originally designed for use with low-bandwidth wired communications such as serial communications and tolerant of long communication latency [9]. Traditional dedicated wired communication protocols for industrial applications have been well designed to meet requirements regarding the deadline.

The IoT needs connectivity for devices and Industrial Wireless Sensor Networks (IWSNs) [10,11] have been emerging as a new means of wireless communications for the IoT. Recently, several protocols for IWSNs such as WirelessHART [12], ISA100.11a [13], and IEEE802.15.4e [14], have been developed and standardized in order to increase the communication reliability in a wireless network with frequent packet loss and big latency under the heavy network utilization. These standards are time-synchronized and assigned communication timings when devices join the wireless network. Both application data packets and network control packets are transmitted according to the assignment.

When polling-based protocols for industrial applications are used over IWSNs, the number of retransmissions of polling queries over IWSN layer should be considered in order to maintain a high reliability of industrial systems. At the same time, their timing of the retransmission should be carefully considered in order to keep the deadline.

Moreover, industrial applications as described above basically gather heterogeneous periodic information from field devices. The transmission frequency of periodic information collection is normally static. At the same time, they also require unpredictable communication for gathering on-demand data or operating devices by a center device within specific end-to-end deadlines [15,16,17]. In general, traditional industrial communication protocols over wired communications take into consideration maintaining the deadline even if the unpredictable communication occurs in the network. For frequent data collection in DA, the wired network load can be as much as ten times less than upper bound [18].

To achieve both heterogeneous periodic data collection with high success ratio and unpredictable on-demand communication within a deadline over IWSNs, we propose a data traffic control scheme for polling-based communication in IWSNs and a scheduler which uniformly distributes network load over slots. Our proposed scheme incorporates multiple heterogeneous periodic data collection schedules in a single schedule. The proposed approach aims to balance network traffic of multiple applications as well as to provide high success data collection. In addition, the schedule gives all nodes fair opportunities to receive a polling query from a central node at a certain interval. It is important to generate a polling query to a node fairly, because the available bandwidth for downward traffic is less than that for upward. This mechanism also enables a center node to generate an unpredictable on-demand request to a node and receive a response within a deadline.

The contribution of this paper is to propose a data traffic control scheme for polling-based communications and verify its performance from viewpoints of end-to-end communication success probability and balanced slot utilization. None of the conventional technologies for IWSNs focus on the problem that an unbalanced network bandwidth between uplink and downlink causes unexpected big latency of unpredictable on-demand communication or decrease of success probability for periodic data collection. Standard IWSN protocols like ISA100.11a typically have a scheduler for allocating network resources such as timeslots to all nodes. Since a network manager of IWSN protocols gathers information about network condition from all nodes, IWSN protocols normally require a large available bandwidth for upward traffic. In addition, the scheduler has to deliver the information to all nodes whenever a new node joins the network or a network topology changes. Therefore, it is difficult to support a polling-based unpredictable on-demand communication within deadline. In contrast, since our scheme only determines when to generate and transmit a packet at a root node for periodic data collection and unpredictable on-demand data collection. It does not need to adjust the schedule because the number of nodes does not change.

The rest of the paper is structured as follows. We first describe assumptions and challenges in Section 2, and Section 3 presents an overview of related work. In Section 4 we propose the data traffic control scheme. Then, we evaluate uniformity and success ratio of collecting periodic data in Section 7. In Section 8, we discuss possibility of providing schedules when an application data traffic is high. Finally, Section 9 concludes the paper and describes future work.

## 2. Assumptions of Our Target System and Challenge

In this section, we provide assumptions of our target system at first, and describe challenges of this paper.

### 2.1. Assumptions


**IIoT application features.** As noted above, our target IoT applications are AMI, DA, PA, and so on. These typical industrial applications normally collect field data from end devices periodically within pre-determined deadlines that include communication latency and internal processing time on both a central node and an end device. In this paper, we assume that the end devices periodically generate data in intervals, and a deadline that the central node should get data from an end device is equal to the next time that an end device generates data within the interval.**polling-based communication.** Our target IoT applications normally use polling-based communication to keep a sequence of processes or simplify network management. In general, there are two different polling schedulings, called monocycle and multicycle polling scheduling.**Monocycle polling scheduling.** Monocycle polling scheduling is the most common and the simplest. In monocycle polling scheduling, a central node uses a single cycle for the polling required for all devices belong to a single group. Figure 1 shows an example of a monocycle polling scheduling. In this case, the deadline (cycle) is $T_{1}$ and three nodes ($n_{1,1},n_{1,2},n_{1,3}$) belong to the group.**Slot assignment for reliable polling communication.** A central node transmits the first queries at $t_{1},t_{2},t_{3}$ to the nodes respectively. Retransmission timings are also allocated in case of communication failure. For example, $t_{4}$ and $t_{7}$ are timing for retries of the first query to $n_{1,1}$. The number of the retries $R_{j}$ for node *j* depends on communication success rate, for example PER (Packet Error Rate) as shown in Equation (Equation 1);
(1)$$1-{P_{PER}}_{j}^{R_{j}}\geq Th_{success},$$
where ${P_{PER}}_{j}$ is an end-to-end PER between a central node and node *j* and $Th_{success}$ as one of system requirements is probability of collecting data from any nodes in a system.**Multicycle polling scheduling.** Multicycle polling scheduling is a set of heterogeneous monocycle polling schedulings. Figure 2 shows an example of multicycle polling patterns. In this example, three applications in a system have different cycles that are denoted by $T_{1}$, $T_{2}$, and $T_{3}$, respectively. The central node transmits the first queries and their retires in each cycle as well as a monocycle polling scheduling. In Figure 2, lower brightness areas represent some consecutive slots of first queries and higher ones do some consecutive slots of their retires. Besides, white areas stand for open slots for unpredictable on-demand request. The width of the areas depends on the schedules. For example, the first lower brightness area of $G_{1}$ includes 3 slots and the second one includes 5 slots.**Wireless communication for IWSNs.** To increase the reliability of wireless networks for industrial applications, Time Slotted Channel Hopping (TSCH) based IWSN protocols such as WirelessHART, ISA100.11a, and IEEE802.15.4e have been developed. They are time-synchronized and assigned communication timings for transmitting packets by a central network manager in order to decrease both interference within the wireless network and interference from other wireless networks using the same radio channels. The communication timing for both downlink traffic from a central node to end devices and uplink traffic from end devices to a central node are assigned. These bandwidths for downward and upward are normally fixed. In addition, the available bandwidth for upward traffic is bigger than that for downward.**Imbalanced bandwidth between upstream and downstream of IWSNs.** Generally speaking, bandwidth of standard IWSNs for the downstream from a center node to end devices is quite less than that for upstream because main traffic for IWSNs is autonomously transmitting sensor data from devices. For example, SmartMesh IP [19] that is based on the 6LoWPAN and IEEE802.15.4e standards provides only one timeslot for downstream traffic from a center node to a device node in every 2 s by default settings, contrary to several timeslots for upstream from the nodes to a center node.**Traffic patterns.** Industrial applications basically gather heterogeneous periodic information from field devices. At the same time, they also require on-demand data communication for additional data collection and operation of devices by a remote server within specific end-to-end deadlines [15]. When the unpredictable on-demand data communication occurs, the unpredictable traffic may be given high priority to over periodical scheduled communication and a central node transmits a polling packet for the on-demand request. Consequently, a periodical polling request from a central node to a node is dropped by the unpredictable on-demand data communication. If an open slot is fortunately assigned when the on-demand request occurs, a central node can get data from a node by an on-demand request without any interruptions of periodic data collection. Therefore, it is better for IWSNs to generate network traffic load for periodical data collection uniformly and also keep available bandwidth for unpredictable traffic at any time.**Traffic control scheme for multiple wireless communications.** Industrial applications require similar requirements for IWSNs, but applications in different domains may use different IWSN protocols. For example, AMI often uses IEEE802.15.4e and PA uses WirelessHART or ISA100.11a. We propose a data traffic control scheme for periodical data collection over IWSNs and our proposal does not completely depend on any specific protocols. Our proposed scheme is a technology that lies between applications and an interface of IWSNs, namely IP (network layer) or MAC (datalink layer) as shown in Figure 3. A center node has the data traffic control function in order to manage all polling traffic.


### 2.2. Challenge

As described above, it is important to realize both collecting multiple periodic data from sensors within deadlines and transmitting unpredictable on-demand packets. Transmitting opportunity for downstream of IWSNs such as ISA100.11a or WirelessHART, is normally lower than that for upstream, even though industrial applications use polling-based protocols that need equal opportunities for downward packets and upward packets. Therefore, one of our challenge is to provide a communication method that controls both downward and upward network load in IWSNs. The communication scheme should also ensure sufficient responsibility of unpredictable requests and stability of periodic data collection. The second challenge of this paper is to provide a schedule which a central node normally generates uniform network load for periodic data collection by a polling-based communication protocol, in order to enable applications to transmit unpredictable packets effectively using the idle time of the IWSNs.

## 3. Related Work

In communication systems and wireless sensor network systems, many studies have been proposed to solve scheduling problems [20,21,22,23,24,25].

In a FieldBus environment, scheduling problems of dynamic or static collecting information flow are to be found in [20,21]. Algorithms capable of solving this problem have to be able to calculate the transmission sequence for all groups. The solution is firstly to decide the primary cycle among multiple cycles. The primary cycle is the shortest period. Then, if there are two polling tasks of different groups at the same time, the group whose polling cycle is $T_{i}$ always has higher priority than group whose polling cycle is $T_{j}$, where $T_{i}\leq T_{j}$. In the papers, authors assume that deadline coincides with the polling cycle, and scheduling problem is to ensure that at least one transmission for each group will occur at least once in the cycle. They do not address features of wireless communication.

In [22], the authors propose TDMA link scheduling algorithms for the purpose of maximizing network throughput. In [23], an end-to-end real-time transmission scheduling over the wirelessHART networks is proposed. Both [22,23] are focus on making conflict-free link level schedules. According to these schedules, end devices can autonomously transmit periodical sensor data to a center node within a deadline according to the schedule but this communication is not polling-based.

In [24], the authors show through analysis and experiments that conflict-free query scheduling has an inherent tradeoff between network throughput and latency. They propose real-time scheduling algorithms for prioritized conflict-free transmission scheduling in order to aim to balance network throughput and latency.

In [25], a new priority-based parallel schedule polling MAC protocol in WSNs combines polling orders with access policies to realize the priority-based scheme and reduce the overhead time through parallel schedule. Sensor nodes can be classified in different clusters and this MAC protocol coordinates all clusters and gives a chance to send data in order. To do this, the node to which a central node wants to transmit a priority packet is given the transmission chance first, and other nodes remain in a sleeping status to save energy.

As will be descried in Section 6, we adopt a GA (genetic algorithm) as a heuristic to derive an optimal schedule [26]. GA is one of probabilistic search algorithms and optimization techniques based on the mechanisms of natural selection and evolution. It has been applied to optimization of complex network management problems such as network load balancing management [27] and routing [28]. Although other heuristic algorithms such as simulated annealing, PSO (particle swarm optimization), or even machine learning can also be adopted, we consider a GA-based algorithm in this paper, which could achieve a reasonable and feasible schedule with practically short computation time on an off-the-shelf computer.

Our work is different from related work in the following aspects. As shown in Table 1, other works focus on how to schedule transmitting queries without conflicts in order to minimize network latency, optimize network throughput, or save energy. However, none of them focuses on the problem that unbalanced network bandwidth between uplink and downlink causes unexpected big latency of unpredictable on-demand communication or decrease of success probability for periodic data collection. We firstly propose a polling-based data traffic control scheme which is available to any IWSN protocols that assign unequal bandwidth to upstream and downstream communication. In addition, we show a scheduling using a network traffic uniformity metric for IWSNs to realize both scheduled periodic data collection at high success ratio and unpredictable on-demand communications with short latency.

## 4. Data Traffic Control Scheme for Periodic Data Collection

In this section, we provide terminologies at first. Then, we present an outline of how our data traffic control scheme for periodic data collection works. We define two kinds of frames and Table 2 summarizes terminologies and notations.
**QueueFrame.** The first kind is the **QueueFrame (QF)**, which consists of $N_{all}$ slots whose length is $∆t_{s}$ as shown in Figure 4. Let $N_{i}$ be the number of nodes which belong to group *i* and Let $\tau_{g}$ be the number of multicycle polling groups in an IWSN. Some nodes may belong to multiple different groups. Then $N_{all}$ in an IWSN is as bellows;
(2)$$N_{all}=\sum_{i=i}^{\tau_{g}}N_{i}.$$We assume that a central node collects data from all nodes at multiple periodic cycle as shown in Figure 2. The length of a QF is $∆T_{QF}\left(=∆t_{s}\times N_{all}\right)$. In our proposal, a central node generates at most $N_{all}$ queries to all nodes in a QF, so that our data collection scheduler gives fair opportunities for all nodes even in a short time span ($∆T_{QF}$). We consider the bandwidth of downward traffic to define polling cycle $∆t_{s}$. For example, a our preliminary experiment showed downward traffic for SmartMesh IP that is based on the 6LoWPAN and IEEE802.15.4e standards is assigned every about 2 seconds by the default setting. $∆t_{s}$ should be longer than the frequency for pre-assigned downlink traffic of IWSN protocols. In this paper, we set $∆t_{s}$ to 3s for simulation setting. The order of queries to nodes are decided by a data collection scheduler. One of the simplest ways to do this is arranging nodes in ascending order of a unique id.**ScheduleFrame.** The second kind is the **ScheduleFrame (SF)**, which consists of QFs, as shown in Figure 5. Let $T_{M}$ be the least common multiple of collection cycle of all groups as shown in Equation (Equation 3). In the interval $T_{M}$, the SF consists of $N_{QF}\left(=T_{M}÷∆T_{QF}\right)$ QFs. Therefore, a SF is the minimum unit of multicycle polling schedule of $N_{all}$ nodes.
(3)$$T_{M}=\mathrm{lcm}\left(T_{i}\right),\mathrm{for}\;i=1,2,\ldots ,\tau_{g}.$$Our data collection scheduler decides all polling orders in order to make network traffic of all applications uniform during the $T_{M}$ and our data traffic control scheme generates polling queries every $∆t_{s}$ according to the schedule.

Below, we give an outline of how our proposal works over any IWSN protocols that assign unequal bandwidth to upstream and downstream communication. In our scheme, we define two kinds of frames to decide all polling orders in both a short time span and a long time span.

Our scheme incorporates multiple groups in a single group as shown in Figure 4. A central node transmits at most $N_{all}$ queries every $∆t_{s}$ according to a sequence of nodes in a QF. This polling behavior is as same as a monocycle polling scheduling. We consider that a central node controls transmitting timing of a polling query so that down stream traffic from a central node is less than pre-assigned downlink traffic of an IWSN protocol.

Then, a scheduler provides a SF that is a multicycle polling schedule during the least common multiple of collection cycle for all groups as shown in Figure 5. In a SF, each QF may assign a different sequence of all nodes. For example, a sequence of all nodes in $QF_{1}$ may be assigned in ascending order and one in $QF_{2}$ may be assigned in descending order.

## 5. Problem Formulation of Data Collection Scheduler

A polling cycle of a group $T_{i}$ consists of $K_{i}$ QFs where
(4)$$\begin{array}{ccc} K_{i} & = & T_{i}/∆T_{QF} \end{array}$$
(5)$$\begin{array}{cc} = & T_{i}/\left(∆t_{s}\times \sum_{k=1}^{\tau_{g}}N_{k}\right) \end{array}$$
(6)$$\begin{array}{cc} = & T_{i}/\left(∆t_{s}\times N_{all}\right). \end{array}$$

Let $C_{i,j}$ be a total number of combinations of selecting QFs which any node *j* in any group *i* uses for data transmit and its retries $R_{i,j}$ is
(7)$$C_{i,j}=K_{i}\times {\sum_{m=1}^{K_{i}}}_{\left(K_{i}-m\right)}C_{min(K_{i}-m,R_{i,j})},$$
where $min(K_{i}-m,R_{i,j})$ is the smallest number of the two arguments.

Total number of combinations of selecting QFs which all nodes in any group *i* use is
(8)$$C_{i,all}=\prod_{j=1}^{N_{i}}C_{i,j}.$$

Then, total number of combinations of selecting QFs for all groups and all nodes $C_{all,all}$ is
(9)$$C_{all,all}=\prod_{i=1}^{\tau_{g}}{\left(C_{i,all}\right)}^{\frac{T_{m}}{T_{i}}}.$$

The number $C_{all,all}$ depends on the number of nodes $N_{all}$ in an IWSN, the number of collection cycles $\tau_{g}$, communication quality, i.e., the number of total communication retries $R_{i,j}$.

On the other hand, maximum number of polling queries for periodic data collection of group *i* during a polling cycle $T_{i}$ is $\sum_{j=1}^{N_{i}}\left(1+R_{i,j}\right)$. Let $Q_{i,x}$ be total polling queries of the *x*th QF of a group *i* during $T_{i}$. When the queries are uniformly generated during $T_{i}$, $Q_{i,x}$ is approximated by the Equation (Equation 10).
(10)$$Q_{i,x}\approx \frac{1}{K_{i}}\sum_{j=1}^{N_{i}}\left(1+R_{i,j}\right)$$

The total polling queries of *x*th QF of all groups is
(11)$$\begin{array}{ccc} Q_{all,x} & = & \sum_{i=1}^{\tau_{g}}Q_{i,x} \end{array}$$
(12)$$\begin{array}{cc} \approx & \sum_{i=1}^{\tau_{g}}\frac{1}{K_{i}}\sum_{j=1}^{N_{i}}\left(1+R_{i,j}\right). \end{array}$$

Then, we find an optimal schedule from $C_{all,all}$ patterns of schedules in order to generate network traffic load uniformly for periodical data collection. The standard deviation of Equation (12) $\sigma (Q_{all,x})$ is a measure of how uniform polling queries are generated during $T_{M}$. Our objective function is
(13)$$\alpha =\min\left\{\sigma \left(Q_{all,x}\right)\right\},where\;1\leq x\leq \frac{T_{M}}{∆t_{s}\times N_{all}}.$$

## 6. GA-Based Slot Assignment Algorithm

Our polling-based data traffic control scheme decides the order of all queries to all nodes according to a schedule that ensures sufficient responsibility of unpredictable requests and stability of periodic data collection. In other words, the slots of queries for periodical data collection and slots for unpredictable on-demand should be uniformly assigned. As we described in Section 3, Genetic Algorithm (GA) is classified one of heuristic functions which can find an optimum solution for above our problem, so that we propose a GA-based slot assignment algorithm.

The process of our GA-based slot assignment algorithm includes generating initial population, selection, mutation, crossover, repair, and evaluation as shown in Figure 6. We describe the specific process of the flow as follows.

### 6.1. Encoding

Our data collection schedule includes two problems for transmitting polling queries. One is how a central node generates polling queries for heterogeneous multiple periodic data collection with high success ratio. Another is how a central node can transmit polling queries for unpredictable on-demand communication within a deadline over IWSNs. We encode a sequence of nodes and kinds of packet (e.g., periodic data collection, retransmissions for periodic data collection, and unpredictable on-demand communication). In our proposal, each individual has two chromosomes. We describe coding chromosome structure at first and then, we present how to generate the initial population as shown in Figure 6.

#### 6.1.1. Chromosome Structure

An example coding chromosome representing a schedule is shown in Figure 7. Each individual has two chromosomes corresponding to a sequence of node numbers and a sequence of slot types. Both chromosomes have $N_{QF}$ QFs which consist of $N_{all}$ slots as shown in Figure 4, respectively.

As the first chromosome, a sequence of node numbers represents transmission orders from a central node to nodes. In this paper, we substitute a simple number like 1,2,3,⋯ as a node number for (Group number, node number) in order to simplify the notation.

As the second chromosome, a sequence of slot types represents kinds of packets that a central node sends to a node at the slot. We define three slot types in this paper. First type “1” represents that a central node transmits an original polling request packet for periodic data collection. Second one “2” represents that a central node sends the retry packet of the original packet to an end node. Final one “3” means that a central node can transmit a query for an unpredictable on-demand collection to any node at the slot.

For example, Figure 7 shows that a central node transmits a request packet to collect periodical application data from node 1 at slot 1, and a retry request packet to node 1 at slot 8 if the central node did not receive the data corresponding to the previous request to node 1. If the central node received a reply from node 1 by slot 8, the central node does not transmit the retry packet to node 1 at slot 8. In addition, a central node may transmit a polling query to any node for an unpredictable on-demand collection at the slot 4 but a central node never transmits a request packet to node 4 for a periodic data collection at the slot.

#### 6.1.2. Initialization

Each Individual of GA population has two chromosomes. At the beginning of the initialization, a data collection scheduler generates $N_{QF}$ QFs such that all nodes in an IWSN are collated at random in order to encode a chromosome that represents an order of node IDs. We think that the random selection provides a search diversity. Our simulation experiment showed that it did not have a big impact on the performance. After that, a chromosome of slot type is assigned. For the slot type chromosome initialization, first of all, the first queries of all nodes to collect data while a periodic cycle are assigned. The assignment of slot type chromosome denotes 1 in the chromosome as shown in Figure 7. Then, the timing of transmitting queries for retries of the collection while a periodic cycle are assigned. The assignment denotes 2 in the slot type chromosome. The number of retries depends on a path quality (end-to-end PER) between a node and a central node. A central node gets the path quality from a network manager in an IWSN, because a network manager perceives network conditions. If a path quality between a node and a central node is 80%, a central node will require at least a retry. Finally, the slot type of other slots that are assigned to the node are 3. “3” means that unpredictable on-demand data traffic from central nodes can be generated at the slot.

### 6.2. Crossover and Mutation

Figure 8 shows an example of mutation, crossover and repair. At the crossover process, we select two individuals randomly as parents and pick a QF up from the node sequence chromosome of the parent, respectively. The QF of the first parent is exchanged with the QF of the second parent in order to generate two new children individuals. Let $N_{indiv}$ be the number of individuals that are generated at the initial phase. Then we have at most $N_{indiv}+2$ kinds of node sequence in $T_{M}$. In other words, there are at most $N_{indiv}+2$ schedules at the moment.

The mutation operation also provides a search diversity by avoiding the local maximums. Mutation operation alters one individual. In our proposal, the alteration of one individual in each mutation operation is considered. A new individual is produced in the same way of the initialization phase. The new one is replaced to an individual of GA population. Mutation operation does not have to be executed before every crossover operation as shown in Figure 6.

### 6.3. Repair

Because each slot type chromosome corresponds to each node sequence chromosome, slot type chromosome of the first parent are also exchanged with one of the second one. However, the exchanges of slot type chromosome is more complicated than the one of node sequence chromosome because “1” (a query for collecting data) appears once during a data collection cycle $T_{i}$ and “2” (a query for retries) should be assigned after the “1”. As shown in Figure 9, slot number 1 and 2 of the slot type chromosome of a child 1 should be repaired after the exchange. In our proposal, the repair follows the 9 cases as shown in Figure 10;
**Case 1-1**:The child chromosome does not need to be repaired at all.**Case 1-2**:After the crossover operation, a new child chromosomes has no “1” during the $T_{i}$. Then, an anterior “3” in the collection cycle should be changed to “1”, and one of the “2” should be changed to “3” if there are more “2” than required. If there is no “3” before the exchanged slot number, the child chromosome does not need to be repaired at all.**Case 1-3**:After the crossover operation, a new child chromosome has no “1” during the $T_{i}$. Then, an anterior “3” in the collection cycle should be changed to “1”. If there are no “3” before the exchanged slot number, the child chromosome does not need to be repaired at all.**Case 2-1**:After this crossover operation, a new child chromosome has additional “1” during the $T_{i}$. Then, an anterior “1” and “2” should be changed to “3”. Also, one of “3” after the new “1” should be “2” in the case that there are fewer “2” than required.**Case 2-2**:The child chromosome does not need to be repaired at all.**Case 2-3**:One of “3” after “1” changes to “2”.**Case 3-1**:After the crossover operation, a new child chromosome has additional “1” during the $T_{i}$. Then, an anterior “1” and “2” should be changed to “3” or a rearward “1” is changed to “3”. Then, one of the “2” should be changed to “3” if there are more “2” than required.**Case 3-2**:one of “2” changes to “3” in case there are more “2” than required.**Case 3-3**:The child chromosome does not need to be repaired at all.

### 6.4. Selection and Fitness Function

After the crossover, mutation, and repair operations, there are $N_{indiv}+2$ individuals. The first $N_{indiv}$ individuals, which have higher fitness values, are selected and transferred to the next generation. The fitness values of chromosomes are calculated from our fitness function (Equation (Equation 13)).

One of our objectives is to uniform the data traffic generation from a central node to nodes. In our proposal, each slot in a QF generates at most one query to collect data and the slot type chromosome represents how many queries are generated in a QF. At the same time, our scheduler tries to maximize communication success rate. When the fitness value is calculated, the communication success rate of all nodes are also calculated and checked. If the success rate of all nodes of a child chromosome does not satisfy our target probability, the individual is discarded.

Finally, when the fitness values of the best 5 slot type chromosomes are not changed after crossover operation, and the best fitness value is less than a target threshold value, we consider the GA operation is convergent. After the maximum iteration, we also consider that the GA operation will not be converged. At that time, we return to the initialization procedure. Our experiments showed that after 100 experiments, the average execution time of slot assignment for 10 nodes was 1544 milliseconds. After the convergence, the best slot type chromosome that is ranked in the top 5 and that is expected the highest average communication success rate of all nodes, is selected as the best schedule.

## 7. Evaluation

### 7.1. Simulation Settings

We implemented our GA-based slot assignment algorithm in C using the gcc compiler and ran all experiments on Mac OS 10.12.6 with an 2.7 GHz Intel Core i5 CPU with 16GB RAM. To evaluate the performance impact of our data traffic control scheme, we performed a set of simulations with 10 end nodes placed statically and randomly in a square field. A central node was placed at the lower left-hand corner of the field, and LOADng that is a routing protocol for low power and lossy networks [29] was applied to create multi-hop routes from all nodes to a central node with a shortest-path metric. A network topology was fixed during a simulation. We also assume that packet loss among neighbors is caused by several factors such as propagation models, signal processing technology, transmitting power, antenna characteristics, and reception sensitivity, but except signal interference from other nodes. We then determined PER of a path quality at random for every node. As shown in Table 3, we set the range of PER to 0–20%. Although the link PER dynamically changes in reality, we assume it is stable and constant in this paper. Evaluation under dynamic environment is left as future work. After creating a network topology, our implementation uses the information and evaluates how a central node generates network traffic load for periodical data collection uniformly.

At first, we randomly divided 10 end nodes to three groups that have different data collection cycles $T_{1}=2$ min, $T_{2}=4$ min, $T_{3}=8$ min, respectively. According to our preliminary experiments, the average number of crossover operations for 100 network topologies was 1071 when our fitness values were converged. Then, we set the maximum number of crossover iteration to be 4000. Table 3 summarizes the details of the other parameter settings.

### 7.2. Simulation Results

As described in Section 5 and Section 6.4, the standard deviation of Equation (12) is measure of how uniform polling queries are generated during $T_{M}$ and Equation (Equation 13) is our fitness function. Therefore, we calculated fitness values by Equation (Equation 13) for evaluation.

First of all, we confirmed that random selection at the initialization step did not have a big impact on the performance. To evaluate this, we created a network topology and then we generated 200 kinds of schedules for a scenario as shown in Table 3. All the results were converged. Then we show one of the detail results as bellow. The fitness values of the top 20 individuals after the GA-based slot assignment are plotted in Figure 11. As shown in Figure 11, minimum standard deviation at the initial random selection where the number of GA operation is 0, was 0.186 and the final one was 0.049. In this case, our proposal generates a schedule for periodic multicycle data collection after 2000 times GA crossover operations. In contrast, the standard deviation of the conventional scheme [25] is 0.49. From uniforming application traffic of periodic data collection point of view, our proposal is superior to the conventional scheme.

The mean value of all slot type during $T_{M}$ is calculated by the Equation (Equation 14). $SI_{ideal}$ is the ideal value when the polling schedule includes all slot assignment of data transmission for all nodes and their retries. As shown in Figure 12, the best schedule in the 20 individuals assigns the ideal number of slots for data transmission and their retries.
(14)$$SI_{ideal}=\frac{1}{N_{QF}\times N_{all}}\times \sum_{i=1}^{\tau_{g}}\left(\frac{T_{M}}{T_{i}}\times SI_{i}\right),$$
where the sum of slot type of all nodes which belong to group *i* is as follows;
$$SI_{i}=N_{i}+2\sum_{j=1}^{N_{i}}R_{i,j}+3\left(K_{i}-N_{i}-\sum_{j=1}^{N_{i}}R_{i,j}\right).$$

In addition, Figure 13 shows E2E communication “3”y of all nodes after the GA-based slot assignment, and average one of all nodes in each individual. The E2E communication “3”y is calculated by the left-side hand of Equation (Equation 1). The abscissa of the graph represents the ID number of individuals, and the ordinate represents the E2E communication “3”ies. The E2E communication “3”ies are over 90% as shown in Figure 13. The average success probability is 97.4% and the lowest one is 95.2% where the 5th individual is selected. The results meet our target requirements, as listed in Table 3.

## 8. Discussion

### 8.1. Possibility of Finding Schedules When Application Data Traffic Is High

In general, the more nodes which belong to the shortest cycle group there are, the central device generates polling requests more frequently during $T_{M}$. In the case that most nodes belong to the shortest cycle group, a scheduler should determine a multicycle scheduling that generates polling transaction uniformly among less options than in the other cases. Then, we additionally evaluate our proposal in the worst case for 10 nodes that 8 nodes in $T_{1}$, 1 node in $T_{2}$ and 1 node in $T_{3}$ respectively. Other conditions listed in Table 3 are the same as the above evaluation. Our proposal scheduler found schedules that meet our target requirements as shown in Figure 14 and Figure 15. The result of the E2E communication “3”y depends not on the grouping ratio but on the network topology.

### 8.2. Possibility for a Scalable IWSN

An IWSN may include over 100 end devices such as sensors or actuators [30], though we evaluated our proposal in Section 7 under a condition that an IWSN includes only 10 end nodes. To evaluate scalability of our proposal scheduler, we increased the number of nodes from 10 to 100. At the same time, we expand polling cycles because our proposal data traffic control scheme has a TDMA-like manner. To be more precise, we enlarged the length of a QF 10 times because the length of a QF is $∆T_{QF}\left(=∆t_{s}\times N_{all}\right)$ as described in Section 4. Accordingly, we also enlarged the length of a SF 10 imes. Simulation conditions are described in Table 4. Figure 16 shows that our proposal also provides a solution which has a scalability that is adaptive to a small scale network to a large scale one with the same uniformity.

### 8.3. Power Consumption

In IIOT systems, many devices such as sensors are powered from batteries. From power consumption point of view, it is important to save energy while devices communicate with others. Due to wireless radio interferences from other systems, PER may increase in IWSNs. Though standard IWSN protocols such as ISA100.11a and WirelessHART increase the reliability of wireless networks by TSCH mechanism, our proposal scheme increases retry queries in harmful network conditions in order to collect data at high success ratio when PER increases. As we described in Section 7, we assume it is stable and constant in this paper. Detailed evaluation is left as future work.

## 9. Conclusions and Future Work

This paper introduced a data traffic control scheme over IWSNs for polling-based data collection from multiple IIoT applications and its data collection scheduling. Our algorithm enables the uniform generation of network traffic load for periodical data collection. Polling request packets from a central node to end nodes can be uniform even if the polling cycles are multiple. This is achieved through a data traffic control scheme that generates a polling query at a fixed interval according to a schedule which is decided to make the occurrence of transmitting a polling request for periodical data collection, its retry in case of wireless communication failure or an interruption of higher priority request, and non operation by a GA-based algorithm. At the same time, our scheduler can give maximum opportunities for retransmitting polling requests and data collection ratio of periodic data packets achieves higher than 90%. We adopted a GA based algorithm to create a schedule, but our data traffic control scheme, of course, allows an algorithm that is not GA-based to decide a schedule.

In this paper, we did not address dynamic adaptation of our scheme to handle dynamic or unexpected changes in application and system requirements. For example, some applications will be installed after the deployment phase. In this case, a scheduler should modify a current schedule. Updating a whole schedule is one of the simplest solutions, but the applications should wait to execute this for at most $T_{M}$. Moreover, in our scenario, all nodes reply a response packet corresponding to a polling request, but an application such as error log monitoring on an end device may transmit more information (including system log data) than that which can be delivered in a reply packet. Therefore, in order to make the data traffic of all applications uniform, a scheduler should consider both uplink traffic and downlink traffic at the schedule creating phase. We plan to tackle these issues as future work.

## Figures and Tables

**Figure 1 sensors-19-00187-f001:**
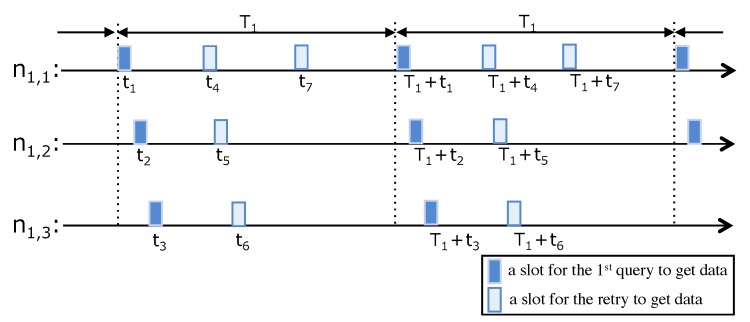
An example of polling schedule in a monocycle polling scheduling.

**Figure 2 sensors-19-00187-f002:**
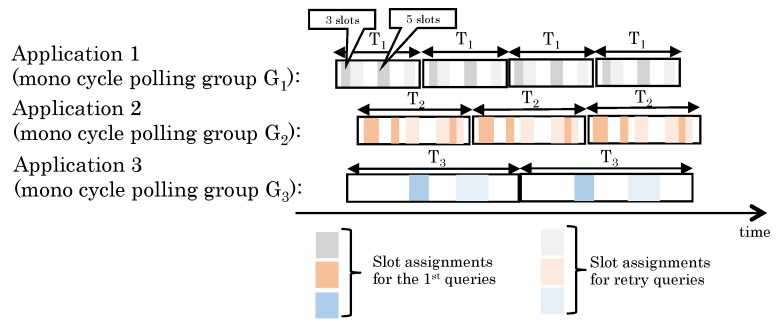
An example of multicycle polling patterns.

**Figure 3 sensors-19-00187-f003:**
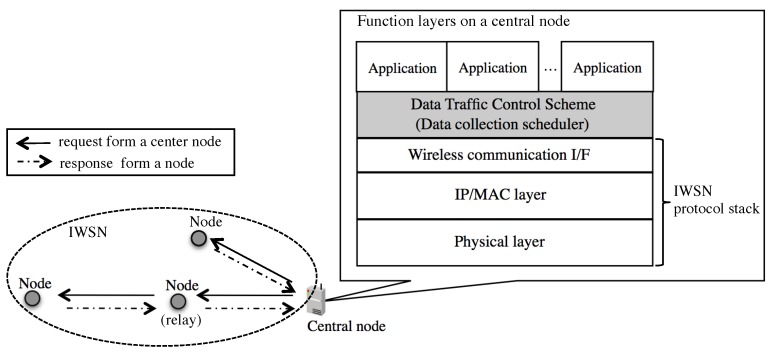
Position of our data traffic control scheme in function layers.

**Figure 4 sensors-19-00187-f004:**
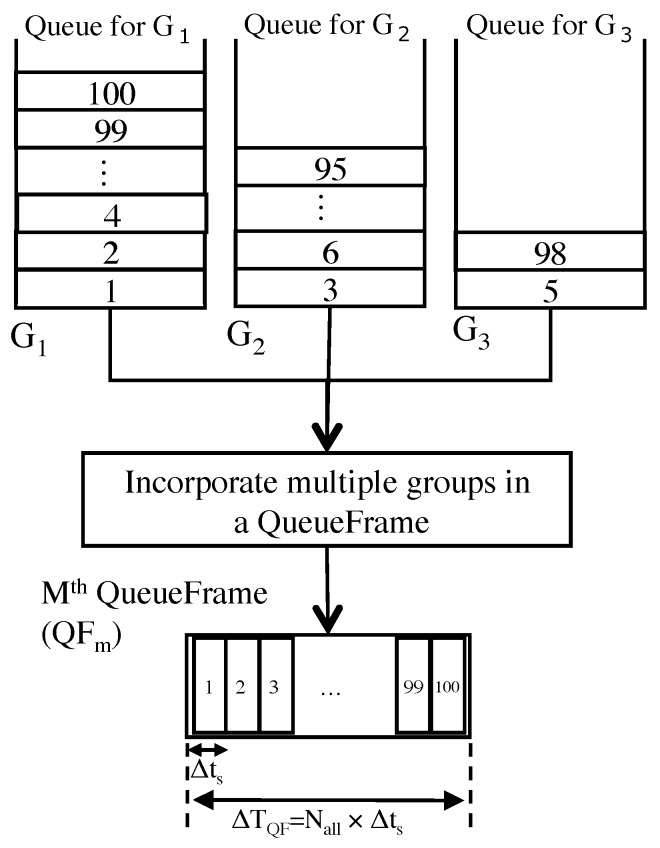
A composition of a QueueFrame (QF).

**Figure 5 sensors-19-00187-f005:**
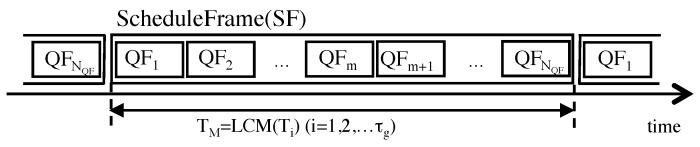
A composition of a ScheduleFrame (SF).

**Figure 6 sensors-19-00187-f006:**
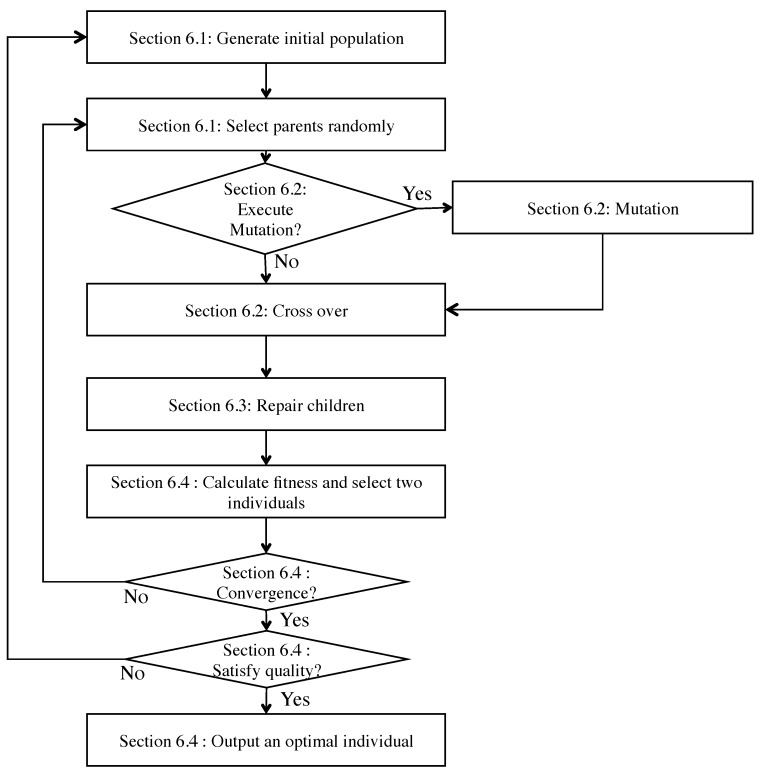
Flowchart of our Genetic Algorithm (GA)-based slot assignment algorithm.

**Figure 7 sensors-19-00187-f007:**
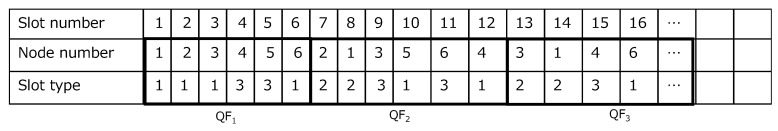
An example of chromosome encoding.

**Figure 8 sensors-19-00187-f008:**
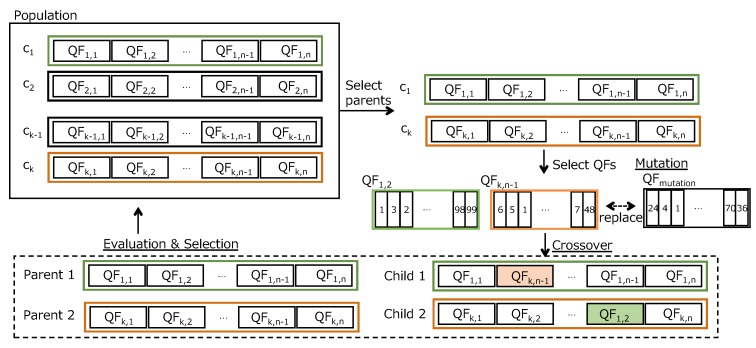
An example of our mutation and selection.

**Figure 9 sensors-19-00187-f009:**
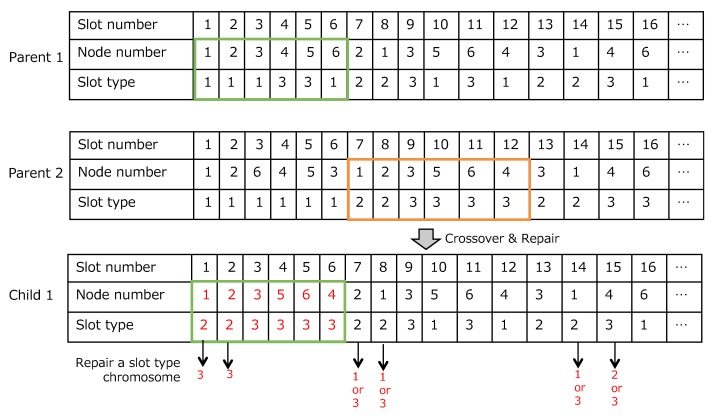
An example of the crossover process.

**Figure 10 sensors-19-00187-f010:**
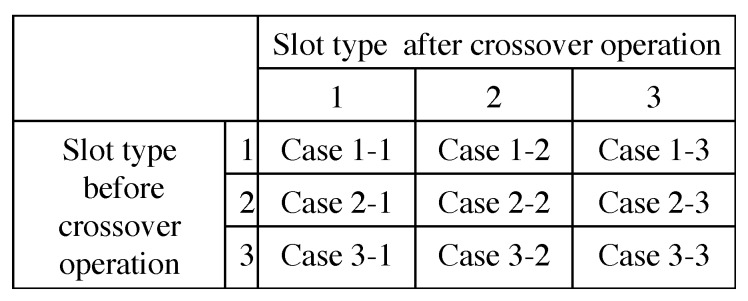
Cases of repairing genes after crossover slot type chromosomes.

**Figure 11 sensors-19-00187-f011:**
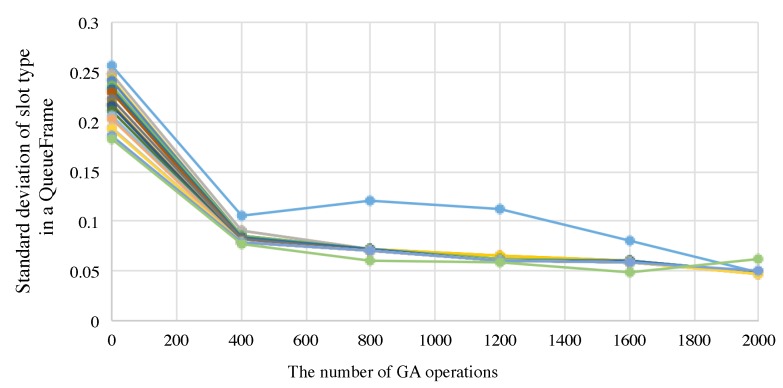
Transition of standard deviation slot type value of the top 20 individuals, where $N_{all}=10$.

**Figure 12 sensors-19-00187-f012:**
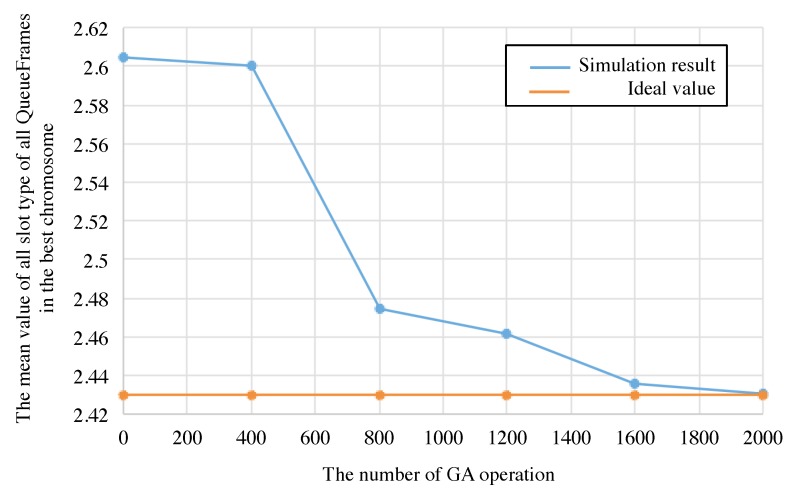
Comparison between the mean value of slot type of all QueueFrames between the best chromosome and ideal value.

**Figure 13 sensors-19-00187-f013:**
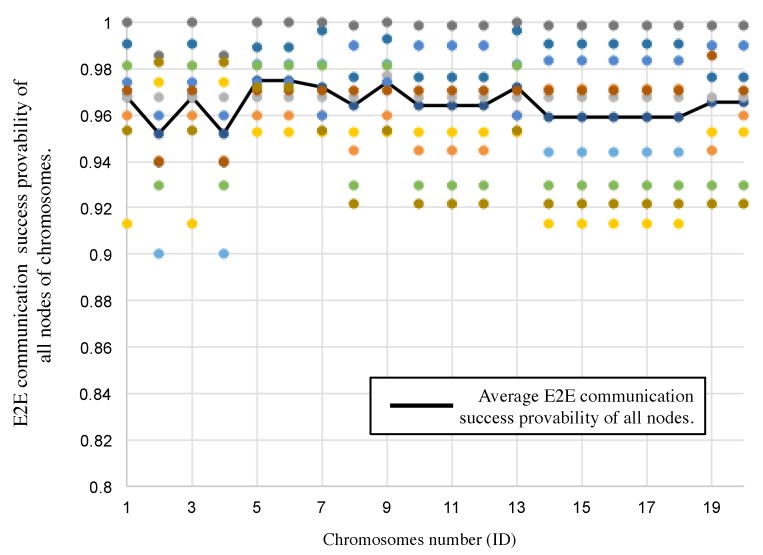
E2E communication “3”y of all nodes in the top 20 individuals and average one of all nodes in each individual.

**Figure 14 sensors-19-00187-f014:**
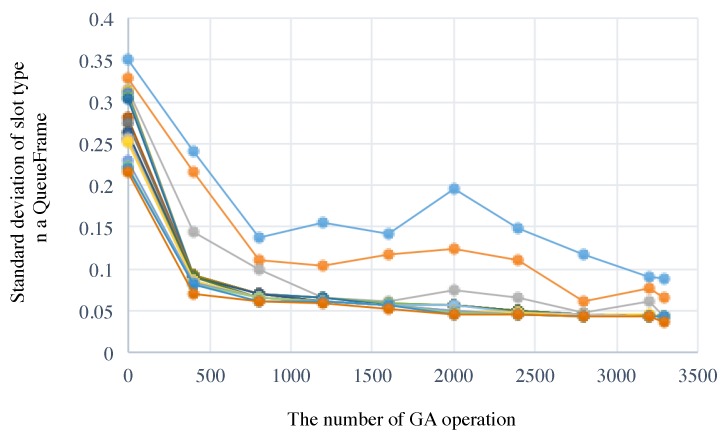
Transition of standard deviation slot type value of the top 20 individuals, where $N_{1}=\;8,$
$N_{2}=1,$ and $N_{3}=1$.

**Figure 15 sensors-19-00187-f015:**
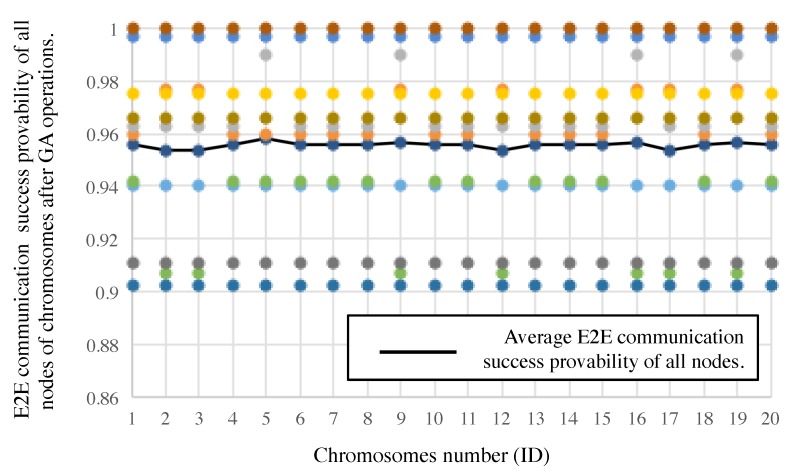
E2E communication "3"y of all nodes in the top 20 individuals and average one of all nodes in each individual, where $N_{1}=8,N_{2}=1,$ and $N_{3}=1$.

**Figure 16 sensors-19-00187-f016:**
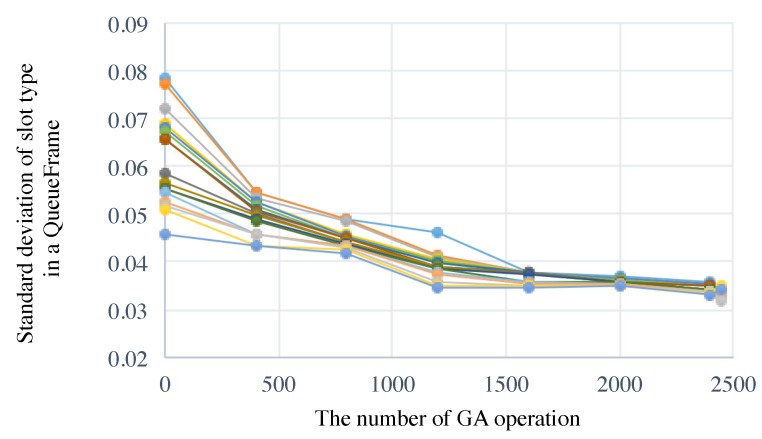
Transition of standard deviation slot type value of the top 20 individuals, where $N_{all}=100$.

**Table 1 sensors-19-00187-t001:** Summary of related work in comparison with our work.

Purpose of Scheduling	Work	Network Type	Communication Technique	Scheduling Type	Unbalanced Bandwidth
Real-time data collection	[20]	Wired	Polling	Conflict-free	Nothing
Multicycle periodic data collection	[21]	Wired	Polling	Conflict-free	Nothing
Optimizing network throughput	[22]	Wireless	Push	Conflict-free	No consideration
Minimizing latency	[23]	Wireless	Push	Conflict-free	No consideration
Minimizing latency and optimizing network throughput	[24]	Wireless	Polling	Prioritized conflict-free	No consideration
Saving energy	[25]	Wireless	Polling	Prioritized	No consideration

**Table 2 sensors-19-00187-t002:** Notation and description.

Notation	Description
$N_{i}$	$N_{i}$ is the number of nodes in group *i*.
$\tau_{g}$	$\tau_{g}$ is the number of multicycle polling groups in an IWSN.
$N_{all}$	$N_{all}$ is the number of total nodes which belong to all groups.
QF	QueueFrame (QF) is the minimum frame which consists of $N_{all}$ slots.
$∆T_{QF}$	$∆T_{QF}$ is The length of a QF. A central node can transmit at most $N_{all}$ polling requests in a QF.
$∆t_{s}$	$∆t_{s}$ is the length of a slot. A central node can transmit a polling request in a slot.
$T_{i}$	$T_{i}$ is a collection data interval of group *i*.
SF	ScheduleFrame is the minimum unit of multicycle polling schedule.
$T_{M}$	$T_{M}$ is a multicycle polling interval.
$N_{QF}$	$N_{QF}$ is the number of QFs in a SF.

**Table 3 sensors-19-00187-t003:** Simulation conditions.

Item	Notation	Value
The number of nodes in an IWSN	$N_{all}$	10 nodes
Data collection cycle 1	$T_{1}$	2 min
Data collection cycle 2	$T_{2}$	4 min
Data collection cycle 3	$T_{3}$	8 min
The least common multiple of collection cycles	$T_{M}$	8 min
The number of slots in a QF	$S_{QF}$	10
Slot length	$∆t_{s}$	3 s
QF length	$∆T_{QF}$	30 ($S_{QF}\times ∆t_{s}$) s
The number of QFs in a chromosomes	$N_{QF}$	16 ($T_{M}÷T_{QF}$)
The number of slots in a chromosomes	$S_{gene}$	160 ($N_{QF}\times S_{QF}$)
The number of individuals	$N_{indiv}$	20
The number of iteration of crossover operation	$I_{max}$	4000 times
Packet error rate	PER	0–20%
MAX retry count	$R_{max}$	2
Threshold of a path quality	$Th_{path}$	90%

**Table 4 sensors-19-00187-t004:** Simulation conditions.

Item	Notation	Value
The number of nodes	$N_{all}$	100 nodes
Data collection cycle 1	$T_{1}$	20 min
Data collection cycle 2	$T_{2}$	40 min
Data collection cycle 3	$T_{3}$	80 min
The least common multiple of collection cycles	$T_{M}$	80 min
The number of slots in a QF	$S_{QF}$	100
Slot length	$∆t_{s}$	3 s
QF length	$∆T_{QF}$	300 ($S_{QF}\times ∆t_{s}$) s
The number of QFs in a chromosomes	$N_{QF}$	160 ($T_{M}÷T_{QF}$)
The number of slots in a chromosomes	$S_{gene}$	1600 ($N_{QF}\times S_{QF}$)
The number of individuals	$N_{indiv}$	20
MAX iteration of crossover operation	$I_{max}$	20,000 times
Packet error rate	PER	0–20%
Max retry count	$R_{max}$	2
Threshold of a path quality	$Th_{path}$	90%

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
